# Kindled emotions: Commemoration and the importance of meaning making, support and recognition

**Huibertha B. Mitima-Verloop** [1,2]*, **Trudy T. M. Mooren** [1,2], **Paul A. Boelen** [1,2]

**1** Department of Clinical Psychology, Utrecht University, Utrecht, The Netherlands, **2** ARQ National Psychotrauma Centre, Diemen, The Netherlands

* b.verloop@arq.org

**Data Availability Statement:** All relevant data are within the paper and its Supporting Information files.

**Funding:** This study was financially supported by the National Committee for 4 and 5 May (The

## Abstract

Commemorative events, organized in the aftermath of war or large-scale violence, can have an emotional impact on those who are attending. We examined several characteristics that might influence this impact. In a quasi-experimental pretest-posttest study, participants ($n$ = 307) watched footage of the broadcast of the Dutch National Commemoration, in which World War II is remembered. A control group of 48 participants watched the commemoration broadcast live on Remembrance Day. They were matched for age, gender, war experience and migration background with 48 participants from the study group who watched the footage, to conduct a comparability check. We found some evidence that watching the footage was comparable to watching the commemoration live on Remembrance Day in terms of emotional response and experience of psychosocial factors. Participants in the footage sample ($n$ = 307) responded with an increase of negative and decrease of positive emotions. Individual characteristics were limitedly related to the emotional response; posttraumatic stress symptom severity predicted increased negative emotions. Experiencing meaning making, support and, to a lesser extent, recognition through commemorating was related to experiencing more positive emotions. The findings indicate these psychosocial factors may buffer the emotional distress elicited by commemoration and contribute to important cognitive and social benefits. Practical implications are discussed.

## Introduction

National days of remembrance, organized in the aftermath of war or large-scale violence, are held in many countries and are often attended by many people. Such commemorative events may have a direct impact on the thoughts, memories, and emotions of participants. Various studies, in different countries and contexts, reported positive emotional responses as a result of commemoration, such as pride, gratitude, feelings of empathy and solidarity [1–3]. Other studies documented primarily negative emotional responses, such as sadness, anxiety, anger, bitterness, and resentment [e.g., 4,5,6]. A recent scoping review revealed that engaging in war commemoration may yield an increase in posttraumatic stress (PTS) and grief reactions [7].

Netherlands)(www.4en5mei.nl), and ARQ Centre of Expertise on War, Persecution and Violence (The Netherlands) (www.arq.org). The funders had no role in study design, data collection and analysis, decision to publish, or preparation of the manuscript.

**Competing interests:** The authors have declared that no competing interests exist.

The present study was designed to enhance knowledge about people's emotional responses to collective war commemoration and factors associated with these responses. This knowledge may ultimately extend our understanding of how and for whom commemoration can be beneficial, contributing to valuable remembrance events.

To date, clinical psychological research on the individual emotional impact of collective commemoration is limited. Mitima-Verloop and colleagues [7] reviewed existing studies and extracted individual characteristics as well as psychosocial factors that might influence the nature and intensity of emotional responses to commemoration. In various studies, female gender [e.g., 2], older age, and war experiences [e.g., 8] were related to more intense emotional responses. Cultural backgrounds are also related to individual responses [e.g., 9], for example because cultural differences may exist about appropriate ways of expressing emotions, especially in public events [5]. Furthermore, the mental health status of people attending commemorations is important. People for whom commemoration yielded an increase in PTS, grief reactions or negative emotions often had a history of mental health problems, mostly related to their own war experiences [e.g., 10, 11]. Even decades after a war, mental health issues can still exist among people who lived through war, prevailing at old age [12, 13]. Also children of those affected can have mental health issues related to the war experiences of their parents (further called second generation affected) [14].

Besides individual characteristics, psychosocial factors might influence individual emotional responses to commemoration [7]. More than a century ago, Durkheim [15] wrote about the emotional and psychosocial consequences of participating in collective rituals on groups, by focusing on shared emotional expression. Durkheim stated that rituals are expected to reactivate emotions associated with the commemorated event. In addition, the emotions expressed by those who had actually experienced the commemorated event could elicit similar feelings in people around them. Durkheim's [15] main premise implied that collective gatherings, regardless of the positive or negative valence of the event, enhance sharing and expressing of emotions, increasing social cohesion, shared beliefs, and positive affect. More recent studies of Paez et al. [16] confirmed this premise from a social psychological perspective, focusing more on individual responses. Their results showed that collective gatherings more strongly reinforced positive affect than that they reduced negative affect among participants. Concurrently, research indicates that the reactivation and spread of negative emotions could install a negative emotional climate among those who are attending [17]. Beristain et al. [5] revealed how emotional expression increased fear, sadness, and anger in participants commemorating the Guatemalan genocide. Thus, studies examining the impact of emotional expression yielded different results, with both positive and negative emotional responses having been reported.

Besides emotional expression, perceived support is another important psychosocial factor to consider when studying the emotional impact of commemoration. Commemorations have the potential to bring people together, to build social support, and to reduce disengagement and isolation [e.g., 18]. Yet, the lack of perceived support during commemorations can also nourish feelings of bitterness [6].

A further factor related to commemoration is recognition or acknowledgement. Collective memorializing fosters acknowledgement of losses, suffering, and the shared public impact of the event that is remembered [19]. Moreover, recognition often reflects societies' desire to account for what has happened and provide justice for those who died [20]. However, how acknowledgement or recognition is perceived by those involved in commemorations might differ and has an impact on the emotional response. For example, the lack of recognition felt by a veteran during VJ Day (Victory over Japan) in the United Kingdom who believed that the British public was ashamed of Far East veterans, made him angry [11]. Furthermore, commemoration and the performance of rituals are related to the construction of meaning [19].

Watkins and Bastian [3] revealed how war commemoration in the United States led individuals to construct meaning by perceiving those who died as costly sacrifices, evoking responses of pride and admiration among citizens.

Lastly, from a clinical psychological perspective, emotional responses after commemoration might be strongly related to personal memories as well. Literature reveals that commemoration can induce intrusions of prior traumatic experiences during war, which are mainly linked to negative emotions such as anxiety [e.g., 21].

## Present study

In the present study, we examined the linkage between engagement in commemoration and emotional responses. In so doing, we considered both positive and negative emotions as orthogonal concepts [22]. Using these terms can lead to oversimplifications and misinterpretations because people often assign a connotative meaning to 'positive' and 'negative' such as good and bad, thinking of them as opposites [23]. In the context of commemoration, experiencing negative emotions does not necessarily mean that the commemoration has a negative impact. For example, Barron et al. [6] described how veterans could (re)experience mentally and physically reminders of the war during commemoration in a safe environment with the support of comrades. Veterans became initially emotionally distressed but also described this as therapeutic. Understanding the context and related factors is essential to interpret the valence of the emotional responses.

This study builds on the review of Mitima-Verloop and colleagues [7], by examining how various individual and psychosocial factors influence emotions after a commemorative event. There is a need to substantiate our knowledge about the potential individual mental impact of commemoration. This is imperative given that commemoration is a widespread practice around the world and is often assumed to assist survivors and society as a whole in coping with disruptive events [24]. In addition, outcomes may guide institutions responsible for organizing commemorative events practically in such a way that it will benefit all individuals.

In this study, a heterogenous group of participants watched footage of the broadcast of the Dutch National Commemoration, in which World War II (WWII) is remembered. Emotions were measured before and after watching the footage. This design allowed us to study various aspects of commemoration in a heterogeneous group of people, while minimizing the differences that are present in a more natural setting, such as what people see or hear based on their physical position during the event.

Our first aim was to investigate if changes in positive and negative emotions could be observed, and which emotions changed mostly while watching the commemoration. The occurrence of emotional changes would confirm our assumption that watching a commemoration has an emotional impact on people, as previous literature has shown for (physically) attending commemorative events [e.g., 5].

Our second aim was to identify individual characteristics associated with emotional responses after commemoration. We considered four demographic variables, namely age, gender, war experience, and cultural background (operationalized as migration background), and two indicators of mental health symptoms, namely PTS symptoms and grief reactions (all assessed prior to watching the footage). Based on previous literature, emotional responses were expected to be relatively stronger among women, individuals with war experiences, and older participants (because of their proximity to WWII), and among individuals with more intense PTS symptoms or grief reactions related to their war experience. Furthermore, we expected participants with a migration background to respond in a less emotional way compared to those without migration background.

Our third aim was to explore to what extent participants reported having experienced emotional expression, support, recognition, meaning making, and personal memories after watching the commemoration. Moreover, we studied the associations of these factors with emotional responses. We expected that the presence of these factors (and the absence in case of personal memories) would predict increased positive and decreased negative emotions after watching the commemoration.

## Materials and methods

### Setting and study design

The study used a quasi-experimental pretest-posttest design, inspired by research using the trauma film paradigm [25]. That paradigm allows to study exposure and responses to psychological trauma, such as negative mood and affect [e.g., 26]. Through presenting footage of potentially distressing and emotional events (e.g., commemoration), analogue emotional response to such events can be studied. In the present study, we used this paradigm to study emotional responses to the Dutch National Commemoration. The Dutch national day of commemoration, Remembrance Day, is held every year on May 4. Besides locally organized events, the National Commemoration takes place in the capital city Amsterdam and is broadcasted live on national television. According to the official memorandum, all civilians and military personnel are commemorated who, in the Kingdom of the Netherlands or elsewhere, died or were killed since the beginning of WWII, in situations of war or in peacekeeping missions. The support for Remembrance Day among Dutch citizens is high and has remained stable over the last twenty years [27].

A fragment of 16:43 minutes of the original television broadcast of the National Commemoration on May 4, 2018 was used. This fragment shows several important rituals, such as the royal family laying the first wreath, the trumpet-signal, singing of the national anthem, two minutes of silence, and those affected by WWII giving their testimony and laying wreaths at the monument. Before (Time 0; T0) and after (Time 1; T1) the film, questionnaires, including questions about emotions, were administered to participants on paper or via an online application. Ethical approval for the study was obtained from the Ethics Review Board of the Faculty of Social Sciences of Utrecht University (FETC18/102).

### Participants

A total of 358 people participated in the study. Inclusion criteria were residing in the Netherlands, having a sufficient level of Dutch language to understand the questions and film, and minimum age of 18 years. Participants who reported not having watched the film with audio ($n$ = 26) or who did not complete the questionnaires concerning emotions before and after the film ($n$ = 25) were excluded from the analyses. Eventually, 307 participants, of which 118 were male (38.4%) and 189 were female (61.6%), were included in the analyses. The age of the participants varied from 18 to 95 years ($M$ = 41.10, $SD$ = 22.57 years). Additional characteristics of participants are presented in the Results section below.

### Procedure

We conducted a pilot study among 13 participants with different backgrounds (in age, gender, war experience or country of origin) to test the procedure, comprehensibility and feasibility of the questionnaires. Based on the remarks by participants, minor refinements were made in some questions. The data of these participants were included in the main analyses. The study was carried out between November 2018 and November 2019 (with the exception of days

around May 4). We purposely included participants with different war experiences (i.e., elderly people who experienced WWII, veterans, refugees from conflict-zones who resettled in the Netherlands, second generation WWII affected, or people with other war-related experiences such as being a reporter in a war zone or being the partner of a traumatized veteran). This enabled us to make comparisons based on war experience, war-related PTS symptoms, and grief symptoms. Participants with war experience were recruited through elderly homes and home-care organizations, Dutch veteran societies, the Dutch Council for Refugees, counsellors from ARQ National Psychotrauma Centre, and the personal network of researchers. Several participants were in treatment for trauma-related problems while participating in the study. People without war experience were recruited via the researchers' own networks, social media, and an online platform of Utrecht University that allowed its students to participate in studies in return for course credits. No other rewards were granted for participation in the study. All participants received an information letter and offered written informed consent prior to the study. Most participants received an e-mail link to participate in the study via a secured online environment. For ethical reasons, most elderly participants, patients in treatment, and refugees participated while in the presence of a researcher, to be able to receive technical or emotional support when needed.

## Comparability check

The emotional response after the film was expected to be comparable to the emotional responses after watching the live broadcast of the National Commemoration on Remembrance Day. The majority of the Dutch population (76%) commemorates on Remembrance Day by following the National Commemoration via (online) television or radio [27], and thus views similar footage as the study group. Comparability was measured with two items in the questionnaire (administered to all participants watching the footage) and through a control-study during the National Commemoration on May 4, 2019. Participants who took part in the control study were recruited via the personal network of the researchers. They were instructed to watch the commemoration via the national broadcast at least between 7:50PM and 8:10PM and fill in a questionnaire directly before and after watching the commemoration. This part of the ceremony covers the same rituals as presented in the footage. One hour before the start of the commemoration, participants received a link to the first part of the online questionnaire via email. Then, at 8:15PM, after the instructed time of watching, a link was sent to the second part of the questionnaire. The sample for this control-study included 48 participants who all confirmed that they had watched the commemoration at least between 7:50PM and 8:10PM. To compare this control group ($n = 48$) with participants who watched the commemoration footage at a random day in the year, we extracted 48 participants from the footage sample ($n = 307$) matched for age, gender, war experience and migration background. Characteristics of both groups are presented in the Results section.

## Measures

*Demographic variables* included (i) gender, (ii) age in years, (iii) level of education (dichotomized as 0 = other than college/university and 1 = college/university), (iv) place of birth and parents' place of birth (dichotomized as 0 = no migration background and 1 = migration background [i.e. at least one parent or oneself born outside of the Netherlands]) and (v) war experience. For war experiences, five categories were included in the questionnaire, namely (a) experienced WWII, (b) experienced war as military personnel, (c) experienced war in country of origin (refugee), (d) other war experiences (including second generation WWII affected),

and (e) no war experience. Answers were collapsed into two categories: 0 = war experience and 1 = no war experience.

*Positive and negative emotions* were measured at T0 and T1 using ten items from the expanded version of the Positive and Negative Affect Scale [PANAS-X; 28]. Five positive (i.e. inspired, happy, proud, concentrating, calm) and five negative (i.e. sad, downhearted, angry, afraid, ashamed) items were chosen, that were expected to represent important emotional responses to commemoration. Participants indicated on a visual analogue scale (VAS; ranging from 0 = not at all to 100 = a lot) how much they felt each emotion at that particular moment. A VAS scale is of most value when examining changes within individuals and can detect even small changes in emotions [29]. Emotions were analysed both as separate items and total scores, calculated by summing the negative and positive items separately. Cronbach's alpha for positive (T0: $\alpha$ = .74; T1: $\alpha$ = .66) and negative emotions (T0: $\alpha$ = .86; T1: $\alpha$ = .81) were satisfactory [30].

*Posttraumatic stress symptoms* related to war experiences were measured at T0 using the Posttraumatic Stress Disorder Checklist [PCL-5; 31]; Dutch translation by Boeschoten et al. [32]. The questionnaire consists of 20 items. Participants indicated how much they had been bothered by these symptoms associated with their own war experience, or the war experience of a close relative, in the past month (e.g., 'Repeated, disturbing dreams of the stressful experience?'). Items are scored on a 5-point Likert scale (0 = not at all to 4 = extremely). Participants without war experience, or without problems related to the war experience of a close relative, did not fill in the questionnaire. Their total score was entered as 0. A cut-off score of ≥31 indicates probable posttraumatic stress disorder [PTSD; 33]. Research has shown good psychometric properties of the scale in veteran samples [34]. Cronbach's alpha in the present study was good, $\alpha$ = .96 [30].

*Grief reactions* related to war-related losses were measured at T0 using the Traumatic Grief Inventory Self-Report [TGI-SR; 35]. The questionnaire consists of 18 items, of which participants indicated how much they had been bothered by these symptoms associated with the death of a loved one during war, in the past month (e.g., 'I had trouble to accept the loss'). Items are scored on a 5-point Likert scale (1 = never to 5 = always). A cut-off score of ≥61 indicates probable prolonged grief disorder [PGD; 36]. Participants who did not have a war-related loss did not fill in this questionnaire, their total score was entered as 18. Research has shown good psychometric properties of the scale [36]. Cronbach's alpha in the present study was .93.

*Psychosocial factors* (i.e. recognition, meaning making, support, expression, and personal memories) were measured at T1 with a self-constructed questionnaire based on previous literature [see 7] and expert consultation. The questionnaire consists of two items per factor, with a total of 10 items. On a 5-point Likert scale (1 = totally not to 5 = extremely) participants scored how much all statements applied to them in response to watching the television broadcast. (I) Recognition was operationalized with two broad statements, to make the items applicable to participants with and without war experiences: 'I experience recognition for (indirect) victims of war' and 'I experience recognition for injustice in the Dutch society'. The inter-item correlation was .35, which falls in the recommended range of .15 - .50 for a reliable scale [37]. (II) Support was operationalized as feelings of social support and feelings of connectedness. The two items representing this factor were 'I feel connected to people around me' and 'I feel supported by people around me'. The inter-item correlation was .59, which is slightly above the recommended range [37]. (III) Meaning making was measured with two items that correspond closely to the items other researchers [e.g., 38] have used to measure meaning in a quantitative manner. The first item deals with sense-making ('I can make sense of, or give meaning to, war experiences'), the second question taps on benefit-finding ('Despite the horrors of war,

I have been able to find any benefit from what happened in the war'). The inter-item correlation was .31. (IV) Expression was measured through the items 'I experience safety and space to allow my emotions and feelings related to war' and 'I can express emotions that I cannot express in everyday situations'. The inter-item correlation was .35. (V) Personal memories were measured with the items 'Personal memories related to war emerge' and 'I have to think about the tragedies that I or my loved ones have experienced during war'. The inter-item correlation was .50.

*Comparability* was measured with two items. The first item, 'Does your emotional response to the television footage you watched match your reaction to commemoration on May 4?', was scored on a five-point Likert scale, ranging from 1 = much less intense to 5 = much more intense, including the option of not applicable because of no experience with commemoration. The second item, 'During the two-minute silence in the television fragment, did you commemorate the same way as you would during commemoration on May 4?', was scored on a five-point Likert scale, ranging from 1 = not at all to 5 = to a very high degree, including 'not applicable' and a comment section to explain the answer.

## Statistical analyses

Statistical analyses were performed using SPSS 27.0 [39]. Missing values of one participant with one missing value on the TGI-SR and another participant with one missing value on the PCL-5, were replaced using person mean imputation [40]. Questionnaires with more than 15% missing values were removed from the analyses. A sensitivity power analysis in G*Power [41] for a linear multiple regression with one dependent and six independent variables, an alpha of .05 and a power of 0.80, indicated that our sample of $n = 307$ (after listwise deletion $n = 281$) was sufficient to detect a small effect size of $f^2 = 0.05$ [42]. Smaller effect sizes were deemed irrelevant for this study.

To perform the comparability check, differences between the footage subsample ($n = 48$) and the control group ($n = 48$) for age, positive and negative emotions, and psychosocial factors were analyzed with mixed ANOVA models and two-tailed independent samples *t*-tests. Outcomes of the two comparability items of the total footage sample ($n = 307$) were reported based on descriptive statistics.

Prior to addressing the aims of the study, a principal axis factor analysis was carried out to investigate whether negative and positive emotions were indeed two separate factors in our dataset. Furthermore, Pearson correlations were reported for relationships between all studied variables, i.e., individual characteristics, mental health symptoms, psychosocial factors, and emotions.

To address our first aim, we used descriptive statistics and two-tailed paired-sample *t*-tests to examine the change in positive and negative emotions. To correct for multiple testing, a significance level of .01 was applied. The assumption of normality was met. Regarding our second and third aim, we conducted linear hierarchical regression analyses based on the residualized change method. A significance level of .05 was applied. Assumptions for normality, linearity, multicollinearity, independent errors, and homoscedasticity were met. Negative emotions at T1 were entered as dependent variable and negative emotions at T0 were added in the first block as control variable. In three separate hierarchical regressions, the second block consisted of individual characteristics, mental health symptoms, and psychosocial factors, respectively. In the fourth regression, all variables that were significant independent variables in the prior models were entered simultaneously. A similar series of four hierarchical regressions was conducted with positive emotions at T1 as dependent variable and positive emotions at T0 as control variable.

**Table 1. Demographic characteristics of participants in subsamples.**

| | | Total footage sample (*n* = 307) | | Footage subsample (*n* = 48) | | Control group (*n* = 48) | |
|---|---|---|---|---|---|---|---|
| | | *n* | % | *n* | % | *n* | % |
| Sex | Male | 118 | 38.4 | 11 | 22.9 | 11 | 22.9 |
| | Female | 189 | 61.6 | 37 | 77.1 | 37 | 77.1 |
| Education | Other than college/university | 185 | 60.3 | 42 | 87.5 | 28 | 58.3 |
| | College/university | 122 | 39.7 | 6 | 12.5 | 20 | 41.7 |
| Background | Migration background[1] | 84 | 27.4 | 48 | 100 | 48 | 100 |
| | No migration background | 223 | 72.6 | 0 | 0 | 0 | 0 |
| War experience | Yes | 121 | 39.4 | 44 | 91.7 | 44 | 91.7 |
| | No | 185 | 60.3 | 4 | 8.3 | 4 | 8.3 |

[1] Self or (one of the) parent(s) born in a country other than the Netherlands.

## Results

### Sample

Characteristics of participants in the total footage sample (*n* = 307), the paired subsample watching the footage (*n* = 48) and the control group watching the commemoration live on Remembrance Day (*n* = 48) are presented in Table 1.

### Comparability check

On average, participants of the control group (*n* = 48), who watched the commemoration live on Remembrance Day, watched for 39 minutes (*SD* = 11.55). This is longer compared to the footage group who saw a film of 16:43 minutes. No significant difference in age was found between the footage subsample (*n* = 48, *M* = 27.71, *SD* = 14.26 years) and the control group (*M* = 29.69, *SD* = 14.65 years), *t*(94) = 0.67, *p* = .50. A mixed ANOVA with negative emotions as within subject variable (measured at T0 vs. T1) and group as between subject variable (control group vs. footage subsample) revealed no significant (group by time) interaction, *F*(1, 93) = 2.11, *p* = .15. Thus, the change in negative emotions was not different between the footage subsample and the control group. A mixed ANOVA with positive emotions as within subject variable and group as between subject variable (control group vs. footage subsample) revealed a significant interaction, *F*(1, 92) = 6.13, *p* = .02. Positive emotions decreased after watching the film in the footage subsample, whereas positive emotions increased after watching the live broadcast of the National Commemoration in the control group, although these changes were not significant (*t*(47) = 1.69, *p* = .10 and *t*(45) = -1.89, *p* = .07 for the footage subsample and control group, respectively).

Concerning the psychosocial factors, independent samples t-tests showed no significant difference between the groups in expression (*t*(94) = 0.99, *p* = .32), meaning making (*t*(94) = 0.98, *p* = .33), recognition (*t*(94) = 0.07, *p* = .95), or personal memories (*t*(94) = 0.27, *p* = .79). This indicates that these factors are experienced in the same way after watching the film of the National Commemoration on a random day in the year, compared to watching the live broadcast of the National commemoration on Remembrance Day. A significant difference was found in support. The footage subsample experienced less support compared to the control group (*M* = 6.10, *SD* = 1.57 vs. *M* = 6.81, *SD* = 1.45, *t*(94) = 2.29, *p* = .02).

In the total footage group (*n* = 307), participants rated their emotional response on average as comparable to their reaction on Remembrance Day (*M* = 2.74, *SD* = 0.85 on a scale with anchors 1 = much less intense and 5 = much more intense). Eleven participants had no

experience with commemoration and 23 participants did not fill in the question. On average, participants commemorated during the two-minute silence in the same way as they would on Remembrance Day ($M = 3.07$, $SD = 1.05$ on a scale with anchors 1 = not at all and 5 = to a very high degree). Eleven participants had no experience with commemoration and 24 participants did not fill in the question.

## Preliminary analyses

The full principal axis factor analysis is presented in the S1 Appendix. Results indicate that positive and negative emotions represented two separate factors in our dataset.

Table 2 presents all Pearson correlations between the predictors and outcome variables that are included in the regression analyses.

On average, participants experienced few PTS symptoms related to war experiences ($M = 6.72$, $SD = 14.28$, min. = 0, max. = 76), 9.8% of the participants scored above the cutoff score indicating possible PTSD. Grief reactions were very low, $M = 20.74$, $SD = 7.56$, min. = 18, max. = 60. No participant scored above the cutoff score indicating possible PGD.

## Emotional change

Table 3 shows all mean scores for negative and positive emotions at T0 and T1, including the significance of change. Overall, negative emotions significantly increased after watching the commemoration with a large effect size, $t(301) = -15.41$, $p < .001$, $d = 0.83$. Sadness increased with a large effect size, whereas the emotions downhearted and angry increased moderate and ashamed and afraid increased to a small extent. With an average score of 166.27 ($SD = 101.68$) at T1, this score was still relatively low, considering a maximum score of 500. Positive emotions significantly decreased over time with a small effect size, $t(287) = 4.90$, $p < .001$, $d = 0.28$, but remained higher compared to the negative emotions at T1 ($M = 284.65$, $SD = 80.40$).

**Table 2. Pearson correlations between outcome variables (positive and negative emotions T1) and predictor variables ($n = 272$).**

|  | 1 | 2 | 3 | 4 | 5 | 6 | 7 | 8 | 9 | 10 | 11 | 12 | 13 | 14 | 15 |
|---|---|---|---|---|---|---|---|---|---|---|---|---|---|---|---|
| 1 T0 positive emotions | 1.00 |  |  |  |  |  |  |  |  |  |  |  |  |  |  |
| 2 T1 positive emotions | .55*** | 1.00 |  |  |  |  |  |  |  |  |  |  |  |  |  |
| 3 T0 negative emotions | -.24*** | -.16** | 1.00 |  |  |  |  |  |  |  |  |  |  |  |  |
| 4 T1 negative emotions | -.14* | -.13* | .57*** | 1.00 |  |  |  |  |  |  |  |  |  |  |  |
| 5 Personal memories | .02 | .09 | .28*** | .26*** | 1.00 |  |  |  |  |  |  |  |  |  |  |
| 6 Support | .23*** | .36*** | -.01 | .09 | .28*** | 1.00 |  |  |  |  |  |  |  |  |  |
| 7 Recognition | .14* | .30*** | -.17** | -.02 | .13* | .45*** | 1.00 |  |  |  |  |  |  |  |  |
| 8 Meaning making | .26*** | .38*** | -.08 | -.04 | .21*** | .45*** | .42*** | 1.00 |  |  |  |  |  |  |  |
| 9 Expression | .08 | .17** | .15* | .32*** | .47*** | .49*** | .39*** | .43*** | 1.00 |  |  |  |  |  |  |
| 10 PTS symptoms | -.21*** | -.14* | .54*** | .41*** | .46*** | -.12 | -.22*** | -.08 | .13* | 1.00 |  |  |  |  |  |
| 11 Grief reactions | .02 | -.02 | .28*** | .17*** | .32*** | .09 | -.05 | .07 | .20*** | .45*** | 1.00 |  |  |  |  |
| 12 Age | .07 | -.01 | .16** | .09 | .45*** | .04 | -.17*** | -.08 | .14* | .34*** | .25*** | 1.00 |  |  |  |
| 13 Gender (1 = female) | -.21*** | -.19** | -.13* | .04 | -.11 | .02 | .10 | -.15* | -.10 | -.23*** | -.17** | -.38*** | 1.00 |  |  |
| 14 Migration (1 = yes) | .05 | .04 | .21*** | .09 | .12 | -.07 | -.16** | -.03 | .02 | .18** | .20** | .05 | -.01 | 1.00 |  |
| 15 War experience (1 = no) | -.02 | .06 | -.36*** | -.13* | -.51*** | -.08 | .21*** | -.04 | -.20*** | -.59*** | -.45*** | -.71*** | .36*** | -.25*** | 1.00 |

*Note.* Listwise deletion.

* $p < .05$

** $p < .01$

*** $p < .001$

**Table 3. Negative and positive emotions before and after watching the National Commemoration, and significance of the change.**

| Emotion | M (SD) T0 | M (SD) T1 | Mchange | t | df | p | Cohen's d |
|---|---|---|---|---|---|---|---|
| Sad | 19.43 (23.38) | 49.50 (28.66) | + 33.47 | -17.40 | 304 | .001 | 1.15 |
| Downhearted | 20.61 (33.33) | 39.80 (27.25) | + 19.19 | -11.94 | 304 | .001 | 0.63 |
| Angry | 14.70 (22.42) | 30.13 (28.22) | + 15.43 | -10.09 | 306 | .001 | 0.61 |
| Afraid | 17.63 (23.65) | 23.01 (23.82) | + 5.38 | -4.67 | 305 | .001 | 0.23 |
| Ashamed | 15.41 (21.75) | 24.79 (26.67) | + 12.40 | -6.12 | 306 | .001 | 0.39 |
| Total negative emotions | 86.59 (90.50) | 166.27 (101.68) | + 79.68 | -15.41 | 301 | .001 | 0.83 |
| Inspired | 51.97 (27.06) | 59.37 (25.04) | + 7.40 | -4.27 | 290 | .001 | 0.28 |
| Happy | 62.30 (22.08) | 41.63 (24.91) | - 20.68 | 13.47 | 304 | .001 | 0.89 |
| Proud | 54.27 (28.67) | 59.18 (26.97) | + 4.91 | -2.93 | 304 | .004 | 0.18 |
| Concentrating | 67.52 (22.67) | 64.49 (22.38) | - 3.03 | 2.27 | 304 | .024 | 0.13 |
| Calm | 72.57 (22.41) | 59.86 (25.57) | - 12.72 | 15.54 | 305 | .001 | 0.53 |
| Total positive emotions | 308.08 (85.16) | 284.65 (80.40) | - 23.43 | 4.90 | 287 | .001 | 0.28 |

Overall, happiness decreased with a large effect size and calmness decreased moderately, concentration did not change significantly, and the emotions inspired and proud increased with a small effect size after the commemoration.

## Associations of individual characteristics with emotions post-commemoration

To achieve our second aim, four linear hierarchical regressions (regression 1–4) were run with positive and negative emotions at T1 consecutively treated as dependent variables. In all analyses, emotions at T0 were entered in block 1 as a control variable.

Independent variables in regression 1 and 2 were four individual characteristics, namely age, gender, migration background and war experience. Negative emotions at T0 explained 32% of the variance in negative emotions at T1; A higher age, female gender and having no war experience predicted more negative emotions at T1 and together explained an additional 2%. Positive emotions at T0 explained 27% of the variance in positive emotions at T1; Male gender and having no war experience predicted more positive emotions at T1 and together explained an additional 2%.

## Associations of mental health symptoms with emotions post-commemoration

Independent variables in regression 3 and 4 were two mental health predictors, namely PTS symptoms and grief reactions. Negative emotions at T0 explained 33% of the variance in negative emotions at T1; PTS symptoms predicted more negative emotions at T1 and explained an additional 1%. Positive emotions at T0 explained 28% of the variance in positive emotions at T1; PTS symptoms or grief reactions did not explain unique variance.

## Associations of psychosocial factors with emotions post-commemoration

Table 4 shows mean scores for psychosocial factors that participants experienced while watching the commemoration. Almost half of the participants experienced recognition 'a lot' to 'extremely' through the commemoration. One third of the participants experienced support 'a lot' to 'extremely'. Meaning making, expression and personal memories were on average experienced moderately.

**Table 4. Mechanisms that are experienced during the commemoration.**

| Experience | min, max | M | SD | n | % of score > 8 (= a lot / extremely) |
|---|---|---|---|---|---|
| Recognition | 2, 10 | 7.01 | 1.80 | 300 | 43.9 |
| Support | 2, 10 | 6.20 | 2.07 | 300 | 29.3 |
| Meaning making | 2, 10 | 5.39 | 1.97 | 299 | 16.0 |
| Expression | 2, 10 | 5.41 | 1.81 | 300 | 13.7 |
| Personal memories | 2, 10 | 5.14 | 2.26 | 301 | 18.9 |

To test our third aim, two linear hierarchical regressions (regression 5–6) were run with positive and negative emotions at T1 consecutively treated as dependent variables. Emotions at T0 were entered in block 1 as a control variable. Independent variables in regression 5 and 6 were five psychosocial predictors, namely recognition, support, meaning making, expression, and personal memories. Negative emotions at T0 explained 32% of the variance in negative emotions at T1; Expression predicted more negative emotions at T1 and explained an additional 6%. Positive emotions at T0 explained 30% of the variance in positive emotions at T1; Recognition, support and meaning making predicted more positive emotions at T1 and together explained an additional 9%.

In the final regression models (7 and 8), all significant variables from the individual characteristics, mental health symptoms and psychosocial factors were included in the equations. Negative emotions at T0, a higher age, female gender, having no war experience, more PTS symptoms and more expression predicted negative emotions at T1. Together these factors explained 44% of the variance in negative emotions at T1, of which 12% was added by the predictors in block 2.

Positive emotions at T0, male gender, having no war experience, more support and more meaning making predicted positive emotions at T1. Together, these factors explained 41% of the variance in positive emotions at T1, of which 11% was added by the predictors in block 2. Outcomes of all regression analyses are summarized in Table 5. Full results of the regression analyses, including estimated regression coefficients, their standard error, p-values, and confidence intervals can be found in the S2 Appendix.

## Discussion

Commemorative events, organized in the aftermath of war or large-scale violence, potentially have an emotional impact on those who are attending. Positive as well as negative emotions can be kindled by these collective events of remembrance. Building on a recent review of Mitima-Verloop et al. [7], we studied several individual characteristics and psychosocial factors that might influence these individual emotional responses. Participants in our study watched a part of the broadcast of the Dutch National Commemoration and filled in a survey before and after the film.

Concerning our first aim, a change in emotions after the commemoration was observed, as expected. This confirmed our assumption that, at least at short term, watching a commemoration has an emotional impact on people, as previous literature has shown for (physically) attending commemorative events [7]. Negative emotions, especially sadness, increased after watching the commemoration and positive emotions, especially happiness, decreased. Two positive emotions however, namely inspiration and pride, increased after watching the commemoration. This corresponds well with the research of Watkins and Bastian [3] among participants of Memorial Day in the United States. Inconsistent with Durkheim's theory [15] and Paez et al. [16], we did not find an overall increase in positive emotions after watching the

**Table 5. Summary of linear hierarchical regression analyses predicting negative and positive emotions at T1.**

| | 1 Individual characteristics | | 2 Mental health symptoms | | 3 Psychosocial factors | | 4 Significant variables | |
|---|---|---|---|---|---|---|---|---|
| | Negative emotions (n = 301), $\beta$ | Positive emotions (n = 287), $\beta$ | Negative emotions (n = 296), $\beta$ | Positive emotions (n = 282), $\beta$ | Negative emotions (n = 293), $\beta$ | Positive emotions (n = 279), $\beta$ | Negative emotions (n = 293), $\beta$ | Positive emotions (n = 278), $\beta$ |
| **Block 1: Emotions** | | | | | | | | |
| T0 negative emotions | .57*** | | .57*** | | .57*** | | .56*** | |
| T0 positive emotions | | .52*** | | .53*** | | .55*** | | .55*** |
| Adjusted $R^2$ | .32*** | .27*** | .33*** | .28*** | .32*** | .30*** | .32*** | .30*** |
| **Block 2: Predictors** | | | | | | | | |
| **T0 negative emotions** | .60*** | | .48*** | | .51*** | | .48*** | |
| **T0 positive emotions** | | .50*** | | .52*** | | .46*** | | .43*** |
| Age | .17* | .01 | | | | | .18** | |
| Gender (1 = female) | .13* | -.14* | | | | | .12* | -.14** |
| Migration (1 = yes) | .03 | -.01 | | | | | | |
| War (1 = no) | .15* | .16* | | | | | .33*** | .12* |
| PTS symptoms | | | .18** | -.04 | | | .28*** | |
| Grief reactions | | | -.03 | .00 | | | | |
| Recognition | | | | | -.01 | .13* | | .10 |
| Support | | | | | -.01 | .15* | | .17** |
| Meaning making | | | | | -.10 | .18** | | .14* |
| Expression | | | | | .30*** | -.08 | .27*** | |
| Personal memories | | | | | .01 | .02 | | |
| Δ Adjusted $R^2$ | .02* | .02* | .01* | .00 | .06*** | .09*** | .12*** | .11*** |

\* $p < .05$

\*\* $p < .01$

\*\*\* $p < .001$

commemoration. This can possibly be explained by the more individual experience of participants, watching the commemoration alone, or the nonsynchronous experience, watching the commemoration on a random day and not at the same time as others during Remembrance Day. The control group of participants on Remembrance Day did experience increased positive emotions and more support compared to the group watching the footage on a random day. This could be interpreted as a support of Durkheim's theory about the linkage between participation in collective gatherings, enhancement of social identity and reinforcement of positive affect.

Findings in relation to our second aim revealed that only some of the individual characteristics were related to the emotional response after watching the commemoration. As expected, older age and female gender were predictive of more negative emotions. Female gender was also predictive of less positive emotions. Partially contrary to our expectations, having no war experience was related to more negative emotions and more positive emotions after watching

the commemoration, while taking initial emotions into account. Moreover, we did not find a relationship between migration background and emotions. These findings indicate that commemorations have the potential to evoke emotions among different attendees, regardless of (cultural) background. It suggests that commemorations rekindle emotions of war affected, but may also have contagious effects by eliciting emotions among those who listen and observe [17]. Notably, and in line with prior studies [e.g., 11], PTS symptoms related to war experiences did predict increased negative emotions after watching, signifying the emotional and possible distressing impact of commemorations for those struggling to cope with their past experiences. Taken together, the studied individual characteristics and mental health symptoms explained only a very small degree of positive and negative emotions after watching, while controlling for initial emotions.

In regard to the third aim, we studied five psychosocial factors, namely recognition, meaning making, support, expression and personal memories, and their relationship with emotional responses after commemoration. On average, all factors were endorsed 'moderately' to 'a lot', indicating that these factors are experienced to a certain degree by most participants while commemorating. The experience of personal memories related to war was not related to positive nor negative emotions. This is in line with our finding that war experience does not lead to increased emotions while commemorating. Frijda [43] stated that, in the context of WWII remembrance, memories are experienced as vivid and do not need a commemoration to come to the fore even 50 years after war. Experiencing more recognition, support and meaning making was associated with more positive emotions after the commemoration, and unrelated to the increase of negative emotions. This suggests that these kind of supporting factors do not reduce emotions such as sadness or downheartedness but can moderate the impact of these feelings. Solomon and Stone [23] describe the phenomenon of mixed feelings, meaning not only experiencing more than one emotion at the same time, but also experiencing emotions that are mixed in itself with both positive and negative connotations. Our results indicate that the degree in which participants experience support and connectedness, recognition for war victims in specific or injustice in general, or can make meaning through commemoration contributes to a positive and valuable commemoration despite feelings of distress.

Inconsistent with Durkheim's theory [15], expression or sharing of emotions was not associated with positive emotions. Only negative emotions were related to expression; participants who felt more openness to express their feelings and thoughts also experienced more negative emotions. An important difference between the present study and Durkheim's work is the collective nature of his theory, explaining the enhancement of common feelings and shared beliefs in contrast to individual emotions. Rimé et al. [17] concluded that sharing an emotional experience may not reduce the emotional upset, but may lead to important cognitive and social benefits. Temporary reactivation of emotions is instrumental in eliciting emotional fusion among participants, which brings them closer together. The resulting empathic process and social integration has emotional, social, and cognitive consequences that seem to buffer the destabilizing effects of emotional events. In addition to social integration, our research suggests that other processes, like support, meaning making and recognition, also buffer the possible distressing effect of commemoration.

It is noteworthy that, in contrast to the individual characteristics, psychosocial factors that develop within social and societal contexts were more strongly associated with emotional responses. Social support, recognition and also meaning making are not intra-personal happenings, but social processes [44]. These results underline the importance of looking at the societal context when studying the impact of collective commemoration on individual responses. Social theories of trauma highlight how memories are formed within a collective context [45] and argue that trauma is a socially mediated attribution, constructed by social

groups and national societies [46]. From a clinical psychological perspective, it is relevant to look at individual emotions. However, individual emotions are formed in a broad social context [44] and especially the societal context needs more attention when studying war related trauma and recovery processes in the field of psychotraumatology [47–49].

## Limitations, strengths, and further studies

Several limitations of this study should be taken into account. First, participants who did not watch the film, for example out of disinterest or overwhelming emotions, were excluded from the analyses. This may have limited the variety in emotional responses. Second, the emotional response was only evaluated directly after watching the commemoration and did not reveal anything about the impact in the hours or days after the commemoration. Third, the results should be interpreted and generalized to more natural settings of commemorations with caution. Participants in the control study differed from the footage study in terms of war experiences. No participants with war experiences were included in the control study. Especially the emotional response of people with severe mental health symptoms related to their war experiences could be different in reality compared to the study setting. The comparability check provided some evidence that watching the film was comparable to watching the commemoration broadcast live on Remembrance Day in terms of emotional response and experience of psychosocial factors. However, watching live on Remembrance Day was experienced as slightly more supporting and elicited more positive emotions. Mimicking a collective gathering merely by watching video footage is limited, especially because the social circumstances are often different from more natural settings [16].

Despite these limitations, the quasi-experimental design of the study is unique in the context of commemoration and the comparability check showed promising results to continue studying the impact of such events in an experimental setting. Another strength of the present study is the use of a comprehensive approach, exploring both negative and positive emotions in relation to various individual and psychosocial factors. This provided deeper insights in the associations and importance of these aspects in relation to each other.

Future research should further examine the potential emotional, cognitive and social benefits of commemoration, including social integration, support, meaning making and recognition. A more qualitative in-depth study of these concepts is needed, for example focusing on what recognition means to those who experienced war, and those who did not. Or how participants describe the meaning they attribute to experiences of war. As our study did not diversify in war experiences and did not find a relation between emotional response and cultural background, further studies might elaborate more on how commemorations are experienced among participants with different war experiences and backgrounds. Furthermore, it is necessary to repeat this study in commemorations with different (political) contexts, such as in Bosnia where history is contested [50], or in Rwanda where some feel forced to commemorate [51], to compare emotional responses and influencing aspects.

## Conclusion

In conclusion, the present research gives deeper insights in what might influence an individual's emotional response to commemoration. It revealed an increase of negative emotions and decrease of positive emotions in the immediate response to commemoration. Individual characteristics seem to contribute to a limited extend to the emotional response, although we should be aware of the possible distress for people with severe war related stress symptoms. Especially clinical practitioners working with war affected patients do well to give attention to the possible stress, but also the supporting aspects, that might be evoked around important

dates of commemoration. Although more research is needed, commemoration might be a form of guided exposure, to bring back memories and emotions in a canalized and secure setting and as such be a part of trauma focused treatments.

The experience of meaning making, support and recognition in commemorating may not reduce negative emotions, but buffer emotional distress and contribute to important cognitive and social benefits. These factors develop within a societal context, which should gain more attention in the field of psychotraumatology. Those involved in organizing commemorative events should aim to create such a setting in which these psychosocial factors can be maximized and thereby contribute to a valuable commemoration.

## Supporting information

**S1 Data.**
(SAV)

**S2 Data. Comparability check.**
(SAV)

**S1 Appendix. Factor analysis.**
(DOCX)

**S2 Appendix. Regression analyses.**
(DOCX)

## Acknowledgments

We would like to thank Ismee Tames for sharing her thoughts and ideas with us while constructing the survey for this study. We thank the students Anna de Vries, Melek Yigid, Paula Corella González, Amber van der Lande and Tamara de Wildt for their contribution in recruitment of participants.

## Author Contributions

**Conceptualization:** Huibertha B. Mitima-Verloop, Trudy T. M. Mooren, Paul A. Boelen.

**Data curation:** Huibertha B. Mitima-Verloop.

**Formal analysis:** Huibertha B. Mitima-Verloop, Paul A. Boelen.

**Funding acquisition:** Trudy T. M. Mooren.

**Investigation:** Huibertha B. Mitima-Verloop, Trudy T. M. Mooren, Paul A. Boelen.

**Methodology:** Huibertha B. Mitima-Verloop, Trudy T. M. Mooren, Paul A. Boelen.

**Project administration:** Huibertha B. Mitima-Verloop, Trudy T. M. Mooren.

**Resources:** Huibertha B. Mitima-Verloop.

**Software:** Huibertha B. Mitima-Verloop.

**Supervision:** Trudy T. M. Mooren, Paul A. Boelen.

**Writing – original draft:** Huibertha B. Mitima-Verloop.

**Writing – review & editing:** Huibertha B. Mitima-Verloop, Trudy T. M. Mooren, Paul A. Boelen.

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
