## [Decision Letter · Decision Letter 0]

18 Jan 2023

PONE-D-22-21025Kindled emotions: Commemoration and the importance of meaning making, support and recognitionPLOS ONE

Dear Dr. Mitima-Verloop,

Thank you for submitting your manuscript to PLOS ONE. After careful consideration, we feel that it has merit but does not fully meet PLOS ONE’s publication criteria as it currently stands. Therefore, we invite you to submit a revised version of the manuscript that addresses the points raised during the review process.

We look forward to receiving your revised manuscript.

Kind regards,

Astrid M. Kamperman

Academic Editor

PLOS ONE

and https://journals.plos.org/plosone/s/file?id=ba62/PLOSOne_formatting_sample_title_authors_affiliations.pdf.

Reviewers' comments:

Reviewer's Responses to Questions

**Comments to the Author**

1. Is the manuscript technically sound, and do the data support the conclusions?

Reviewer #1: Yes

Reviewer #2: Yes

2. Has the statistical analysis been performed appropriately and rigorously? 

Reviewer #1: Yes

Reviewer #2: Yes

3. Have the authors made all data underlying the findings in their manuscript fully available?

Reviewer #1: Yes

Reviewer #2: Yes

4. Is the manuscript presented in an intelligible fashion and written in standard English?

Reviewer #1: Yes

Reviewer #2: Yes

5. Review Comments to the Author

Reviewer #1: The study examined the changes in positive and negative emotions after watching the commemoration footage of World War II and the association between emotions post-commemoration and individual characteristics (e.g., age, gender, war experience, migration status, posttraumatic stress symptoms, grief reactions), as well as psychosocial factors (e.g., recognition, meaning making, support, expression, personal memories). This is an interesting and important topic, and the authors properly recognize the limitations of their study and offer alternative explanations for their findings. However, I have several comments that if addressed, would improve the quality of this manuscript:

I understand that the authors are trying to describe the context of Covid-19 in the introduction, but the first paragraph needs to include research questions (or at the very least general aims) as well as an overarching rationale for the study in order to provide clarity to the readers on what the article is about. The reader does not know the research questions until p. 6 of the paper.

It is unclear how the second item of the recognition subscale of the psychosocial factors (i.e., “I experience recognition for injustice in the Dutch society.” in response to watching the commemoration) measures the construct described in the literature review (e.g, the shared public impact of war) or in the ‘Discussion’ section on page 25 (e.g., recognition for their own suffering or for those affected by war).

It is unclear from the paragraph describing ‘Procedure’ and ‘Measures’ what is considered as war experience. According to ‘Procedure’, second generation WWII-affected is also included as having war experience?

Sample characteristics should not be in the ‘Results’ section - it should be combined with the ‘Participants’ paragraph - otherwise the reader is left with many questions about the sample for several pages.

The authors omitted gender as a significant predictor of negative affect on page 19 when explaining the final regression model in writing. Also, the ‘Results’ section should include the interpretation of the results. For example, female gender was related to lower levels of positive affect after watching the commemoration on footage.

On page 16, when describing the preliminary analyses, the authors need to communicate clearly the direction of the correlation of significant variables. For example, older age was correlated with lower levels of recognition.

On Page 17 in the second paragraph, the authors mention Rime’s article on positive emotions, however, it is unclear what its finding is in the literature review. On page 4, the author only indicates that “the reactivation and spread of negative emotions could install a negative emotional climate among those who are attending.” The authors do not introduce Rime’s findings on positive emotions until page 25.

On page 24 in the first sentence, there appears to be a grammatical error - individual characteristics were only “limited related to” the emotional response.

In the ‘Present study’ section on page 7, the authors misrepresent the third aim by stating the association of these factors with “emotional change,” whereas the study examines the association of psychosocial factors with “emotional responses after commemoration” as stated on page 24.

On page 25 in the second sentence, it is unclear what the authors mean by “provide a source of strength next to having these (negative) feelings.”

In future research, it would be interesting to examine the cognitive and social benefits mentioned on page 25 including social integration and emotional synchrony among commemoration participants.

On page 26 in the last sentence, there appears to be a grammatical error - recognition seem (to be) important aspects.

On page 27, the authors recommend the study to be repeated in different contexts to explore other emotional responses. Could the authors elaborate on what they mean? Also, in ‘Conclusion’ in the first sentence, the use of the word “determines” may be oversimplifying the results. In the last paragraph in ‘Conclusion’, the authors used “meaning-making” instead of “meaning making.”

Reviewer #2: A very competent study. I would have liked to have seen more discussion of the social theories of trauma included and also the discussion of Durkheim's work expanded. Durkheim, after all is known for the collective nature of his theories of memory and its transmission. Your paper focuses on individual responses to collective experience, while Durkheim, I believe, was more concerned with the social nature of experience. Individual-based explanation and the field of psychology itself was the target of his critique of the social science of his time.

6. PLOS authors have the option to publish the peer review history of their article (what does this mean?). If published, this will include your full peer review and any attached files.

Reviewer #1: No

Reviewer #2: No

---

## [Author Response · Author response to Decision Letter 0]

17 Feb 2023

Reviewer #1

The study examined the changes in positive and negative emotions after watching the commemoration footage of World War II and the association between emotions post-commemoration and individual characteristics (e.g., age, gender, war experience, migration status, posttraumatic stress symptoms, grief reactions), as well as psychosocial factors (e.g., recognition, meaning making, support, expression, personal memories). This is an interesting and important topic, and the authors properly recognize the limitations of their study and offer alternative explanations for their findings. 

We much appreciate this positive reflection.

However, I have several comments that if addressed, would improve the quality of this manuscript: I understand that the authors are trying to describe the context of Covid-19 in the introduction, but the first paragraph needs to include research questions (or at the very least general aims) as well as an overarching rationale for the study in order to provide clarity to the readers on what the article is about. The reader does not know the research questions until p. 6 of the paper.

The first paragraph describes how countries organize commemoration in the aftermath of war, and the diversity in emotional impact that these events can have on individuals. We did not understand the reviewers comment on Covid-19. We did not try to describe the context of Covid-19, commemorating a pandemic is different from commemoration in a context of war and large-scale violence, on which we based our research. We tried to make this purpose and the aim of the study more clear from the first paragraph by modifying the last sentence of the first paragraph as follows (p. 3): “The present study was designed to enhance knowledge about people’s emotional responses to collective war commemoration and factors associated with these responses. This knowledge may ultimately extend our understanding of how and for whom commemoration can be beneficial, contributing to valuable remembrance events.”

It is unclear how the second item of the recognition subscale of the psychosocial factors (i.e., “I experience recognition for injustice in the Dutch society.” in response to watching the commemoration) measures the construct described in the literature review (e.g, the shared public impact of war) or in the ‘Discussion’ section on page 25 (e.g., recognition for their own suffering or for those affected by war).

Because many participants in the study did not have experience with war, we tried to state both items of the recognition subscale in a very general way, so not only those affected by war but also others could relate to it. We included the phrasing of ‘recognition for injustice’ to complement the item of ‘recognition for (indirect) victims of war’ based on the literature related to recognition described in our previous scoping review (Mitima-Verloop et al., 2020). Literature revealed that participants who did not experience recognition through commemoration, often experienced feelings of neglect and injustice (for example Oushakine, 2006, referred to in Mitima-Verloop et al., 2020). To make this more clear to the reader, we added the aspect of injustice in a sentence on page 5: “Moreover, recognition often reflects societies' desire to account for what has happened and provide justice for those who died [20].”

Furthermore, we changed the sentence on page 28 in the discussion, to make it more aligned with the scale items: “… recognition for war victims in specific or injustice in general”

We do admit that the two questions are limited in trying to capture the broad concept of recognition. This was therefore described as relevant to further study (p. 30: “A more qualitative in-depth study of these concepts is needed, for example focusing on what recognition means to those who experienced war, and those who did not.”)

It is unclear from the paragraph describing ‘Procedure’ and ‘Measures’ what is considered as war experience. According to ‘Procedure’, second generation WWII-affected is also included as having war experience?

We indeed included people who were indirectly affected by WWII as having war experience. We further specified this type of war experience in the introduction (page 4: “Also children of those affected can have mental health issues related to the war experiences of their parents (further called second generation affected)”. Furthermore, we included all categories of ‘war experience’ as were part of the questionnaire on page 11: “For war experience, five categories were included in the questionnaire, namely (a) experienced WWII, (b) experienced war as military personnel, (c) experienced war in country of origin (refugee), (d) other war experiences (including second generation WWII affected), and (e) no war experience. Answers were collapsed into two categories: 0 = war experience and 1 = no war experience.”

Sample characteristics should not be in the ‘Results’ section - it should be combined with the ‘Participants’ paragraph - otherwise the reader is left with many questions about the sample for several pages.

We understand this commentary of the reviewer. However, we did include these sample characteristics in the results section because the table includes several aspects (such as ‘war experience’, ‘migration background’ and the division in footage sample and control group), that are explained in later paragraphs in the methods section, after the paragraph of participants. By adding this information in the ‘participants’ paragraph, the reader would not yet understand how the aspects in the table are defined or measured. In our experience, it is not uncommon that detailed information about the sample is presented in the results section.

The authors omitted gender as a significant predictor of negative affect on page 19 when explaining the final regression model in writing. 

We thank the reviewer for this comment. This was a mistake, gender is added as a significant predictor on page 22. 

Also, the ‘Results’ section should include the interpretation of the results. For example, female gender was related to lower levels of positive affect after watching the commemoration on footage.

We have included interpretations of the regression analyses by adding the following sentences: “A higher age, female gender and having no war experience predicted more negative emotions at T1” (p. 21); “Male gender and having no war experience predicted more positive emotions at T1 .” (p. 21); “PTS symptoms predicted more negative emotions at T1” (p. 21); “Expression predicted more negative emotions at T1 and explained an additional 6%” (p. 22); “Recognition, support and meaning making predicted more positive emotions at T1 and together explained an additional 9%” (p. 22). “Negative emotions at T0, a higher age, female gender, having no war experience, more PTS symptoms and more expression predicted negative emotions at T1” (p. 22). “Positive emotions at T0, male gender, having no war experience, more support and more meaning making predicted positive emotions at T1” (p. 23)

On page 16, when describing the preliminary analyses, the authors need to communicate clearly the direction of the correlation of significant variables. For example, older age was correlated with lower levels of recognition.

We thank the reviewer for bringing this up. We seriously considered this suggestion. However, reporting the direction of the correlations would yield a lot of extra text and, in our view, would be at the expense of the readability of the text. Furthermore, the correlations are not directly part of our research questions. The correlation matrix is only added to better interpret the findings of the regression analyses.

On Page 17 in the second paragraph, the authors mention Rime’s article on positive emotions, however, it is unclear what its finding is in the literature review. On page 4, the author only indicates that “the reactivation and spread of negative emotions could install a negative emotional climate among those who are attending.” The authors do not introduce Rime’s findings on positive emotions until page 25.

We did indeed not mention any study of Rimé linked to positive emotions before page 26. The book chapter of Rimé et al. that is referred to includes a description of different studies with varying outcomes. To keep the message clear to the reader and in line with the introduction, we omitted the reference to Rimé et al. on page 26. The nuanced conclusion of Rimé et al., based on the various studies they describe, is presented on page 28.

On page 24 in the first sentence, there appears to be a grammatical error - individual characteristics were only “limited related to” the emotional response.

We changed the sentence to “individual characteristics were only to a limited extend related to …” (p. 27).

In the ‘Present study’ section on page 7, the authors misrepresent the third aim by stating the association of these factors with “emotional change,” whereas the study examines the association of psychosocial factors with “emotional responses after commemoration” as stated on page 24.

We studied indeed the emotional responses after commemoration, while taking initial emotions into account. We changed the sentence on page 7 as suggested by the author. The same comment also applies to our second aim so also here we made a change to ‘emotional responses after commemoration’, instead of ‘emotional change’ (p.7).

On page 25 in the second sentence, it is unclear what the authors mean by “provide a source of strength next to having these (negative) feelings.”

To make the meaning of this sentence more clear to the reader, we changed it to “This suggests that these kind of supporting factors do not reduce emotions such as sadness or downheartedness but can moderate the impact of these feelings” (p. 28)

In future research, it would be interesting to examine the cognitive and social benefits mentioned on page 25 including social integration and emotional synchrony among commemoration participants.

We have included this suggestion in the paragraph about future research (p. 30): “Future research should further examine the potential emotional, cognitive and social benefits of commemoration, including social integration, support, meaning making and recognition. A more qualitative in-depth study of these concepts is needed … ”

On page 26 in the last sentence, there appears to be a grammatical error - recognition seem (to be) important aspects.

By modifying some parts of the discussion, based on comments of reviewer 2, this sentence changed as well.

On page 27, the authors recommend the study to be repeated in different contexts to explore other emotional responses. Could the authors elaborate on what they mean? 

We have elaborated on this aspect by editing the following sentence on page 31: “Furthermore, it is necessary to repeat this study in commemoration with different (political) contexts, such as in Bosnia where history is contested [50], or in Rwanda where some feel forced to commemorate [51], to compare emotional responses and influencing aspects.” 

Also, in ‘Conclusion’ in the first sentence, the use of the word “determines” may be oversimplifying the results. 

Instead of ‘determines’, we have now used the words ‘might influence’, to do justice to the nuances of the research results (p. 31).

In the last paragraph in ‘Conclusion’, the authors used “meaning-making” instead of “meaning making.”

The word meaning making is changed as suggested. 

We thank the reviewer for their helpful review.

Reviewer #2

A very competent study. I would have liked to have seen more discussion of the social theories of trauma included and also the discussion of Durkheim's work expanded. Durkheim, after all is known for the collective nature of his theories of memory and its transmission. Your paper focuses on individual responses to collective experience, while Durkheim, I believe, was more concerned with the social nature of experience. Individual-based explanation and the field of psychology itself was the target of his critique of the social science of his time.

We thank the reviewer for the positive feedback and the important note on Durkheim’s theory. As the reviewer rightly comments, Durkheim focused more on the group response, and how collective gatherings impacted group belonging. The studies of Rimé and Paez that build on Durkheims theory meanwhile take a social psychological approach, addressing individual responses within the social context. To clarify this distinction, we have modified the text in the introduction page 4: “(..) Durkheim [15] wrote about the emotional and psychosocial consequences of participating in collective rituals on groups by focusing on shared emotional expression. (…) More recent studies of Paez et al. [16] confirmed this premise from a social psychological perspective, focusing more on individual responses. Their results showed that collective gatherings reinforced more positive affect than reducing negative affect among participants.

In the discussion, we emphasized the difference in Durkheim’s theory and our focus on individual responses (p. 28): “An important difference between the present study and Durkheim’s work is the collective nature of his theory, explaining the enhancement of common feelings and shared beliefs in contrast to individual emotions.”

Furthermore, we added a paragraph in which we discuss our results in connection to social theories of trauma and the need for further focus on the societal context in psychotraumatology, page 29: “It is noteworthy that, in contrast to the individual characteristics, psychosocial factors that develop within social and societal contexts were more strongly associated with emotional responses. Social support, recognition and also meaning making are not intra-personal happenings, but social processes [44]. These results underline the importance of looking at the societal context when studying the impact of collective commemoration on individual responses. Social theories of trauma highlight how memories are formed within a collective context [45] and argue that trauma is a socially mediated attribution, constructed by social groups and national societies [46]. From a clinical psychological perspective, it is relevant to look at individual emotions. However, individual emotions are formed in a broad social context [44] and especially the societal context needs more attention when studying war related trauma and recovery processes in the field of psychotraumatology [47-49].

Lastly, this point is also stressed in the conclusions on page 31: “These factors develop within a societal context, which should gain more attention in the field of psychotraumatology.”

We thank the reviewer for the valuable suggestions for improvement of the paper.

---

## [Decision Letter · Decision Letter 1]

21 Mar 2023

PONE-D-22-21025R1Kindled emotions: Commemoration and the importance of meaning making, support and recognitionPLOS ONE

Dear Dr. Mitima-Verloop,

Thank you for submitting your manuscript to PLOS ONE. After careful consideration, we feel that it has merit but does not fully meet PLOS ONE’s publication criteria as it currently stands. Therefore, we invite you to submit a revised version of the manuscript that addresses the points raised during the review process.

We look forward to receiving your revised manuscript.

Kind regards,

Astrid M. Kamperman

Academic Editor

PLOS ONE

Journal Requirements:

Reviewers' comments:

Reviewer's Responses to Questions

**Comments to the Author**

1. If the authors have adequately addressed your comments raised in a previous round of review and you feel that this manuscript is now acceptable for publication, you may indicate that here to bypass the “Comments to the Author” section, enter your conflict of interest statement in the “Confidential to Editor” section, and submit your "Accept" recommendation.

Reviewer #1: All comments have been addressed

2. Is the manuscript technically sound, and do the data support the conclusions?

Reviewer #1: Yes

3. Has the statistical analysis been performed appropriately and rigorously? 

Reviewer #1: Yes

4. Have the authors made all data underlying the findings in their manuscript fully available?

Reviewer #1: Yes

5. Is the manuscript presented in an intelligible fashion and written in standard English?

Reviewer #1: Yes

6. Review Comments to the Author

Reviewer #1: Thank you for your comprehensive responses to the reviewer’s comments. I have two more comments that if addressed, would further improve the quality of this manuscript:

On page 5 in the fifth sentence, there appears to be a grammatical error - “the British public was ashamed” instead of “the British public were.”

On page 27 in the first sentence, it would be more clear to state “Findings in relation to our second aim revealed that only some of the individual characteristics were related to the emotional response after watching the commemoration.”

7. PLOS authors have the option to publish the peer review history of their article (what does this mean?). If published, this will include your full peer review and any attached files.

Reviewer #1: **Yes: **Christine So

---

## [Author Response · Author response to Decision Letter 1]

31 Mar 2023

Dear Dr. Kamperman,

We hereby submit a revised version of our paper “Kindled emotions: Commemoration and the importance of meaning making, support and recognition” for publication in PLOS ONE. We thank the reviewer for her positive feedback and we have made the changes that were suggested:

On page 5, we have corrected the grammatical error as suggested (“the British public was ashamed”). On page 27, we have changed the first sentence as suggested by the reviewer: “Findings in relation to our second aim revealed that only some of the individual characteristics were related to the emotional response after watching the commemoration.”

Furthermore, we have reviewed our reference list to ensure that it is complete and correct. Some small changes, all related to reference style, were made (see Revised Manuscript with track changes). 

We hope that the paper is now ready for publication. We look forward to receiving your response.

Kind regards, also on behalf of the other authors,

H.B. Mitima - Verloop

ARQ National Psychotrauma Centre, Diemen, The Netherlands

---

## [Editor Report · Decision Letter 2]

10 Apr 2023

Kindled emotions: Commemoration and the importance of meaning making, support and recognition

PONE-D-22-21025R2

Dear Dr. Mitima-Verloop,

We’re pleased to inform you that your manuscript has been judged scientifically suitable for publication and will be formally accepted for publication once it meets all outstanding technical requirements.

Kind regards,

Astrid M. Kamperman

Academic Editor

PLOS ONE
---

## [Editor Report · Acceptance letter]

14 Apr 2023

PONE-D-22-21025R2 

Kindled emotions: Commemoration and the importance of meaning making, support and recognition 

Dear Dr. Mitima-Verloop:

I'm pleased to inform you that your manuscript has been deemed suitable for publication in PLOS ONE. Congratulations! Your manuscript is now with our production department. 

Kind regards, 

on behalf of

Dr. Astrid M. Kamperman 

Academic Editor

PLOS ONE